

# Trait-based predation suitability offers insight into effects of changing prey communities

Benjamin Weigel[1,2] and Erik Bonsdorff[1]

[1] Environmental and Marine Biology, Faculty of Science and Engineering, Åbo Akademi University, Turku, Finland
[2] Research Centre for Ecological Change, Organismal and Evolutionary Biology Research Programme, University of Helsinki, Helsinki, Finland

Corresponding author
Benjamin Weigel,
benjamin.weigel@helsinki.fi

## ABSTRACT

Increasing environmental pressures and human impacts are reshaping community structures and species interactions throughout all trophic levels. The morphological and behavioural characteristics of species communities contain key ecological information on why prey species appear attractive to predators but are rarely applied when exploring predator-prey (PP) relationships. Expanding our knowledge on how changing prey communities can alter the food resource suitability (RS) for predators is vital for understanding PP dynamics in changing ecosystems. Detailed predator diet data are commonly restricted to commercially important species and often not available over long temporal scales. To find out whether structural changes of prey communities impact the food RS for predator communities over space and time, we apply a novel framework to describe and interpret changes in predator diet-suitability based on predation-relevant traits of prey. We use information on described feeding links from the literature to compile the prey spectrum for each predator and subsequently translate the prey-species into a prey-trait spectrum. For each predator, we then calculate a frequency-based prey-trait affinity score and relate it to the available food resource pool, the community weighted means of prey traits, resulting in a prey-suitability measure. We aim to reveal whether a described multi-decadal change in the community structure of zoobenthos had an impact on the food suitability for the benthic-feeding fish in a coastal system of the Baltic Sea. We assess the direction of change in resource quality from the perspective of benthic-feeding fish and describe predator-specific responses to examine which species are likely to profit or be disadvantaged by changes in their prey spectrum. Furthermore, we test the relationship between functional diversity of prey communities and food suitability for predators, and whether predation linkage-structures are affected through prey community-changes. Our results show that changes in zoobenthic communities had a positive effect on the food suitability for most benthic-feeding fish, implying more suitable food resources. Species-specific responses of predators suggest varying plasticity to cope with prey assemblages of different trait compositions. Additionally, the functional diversity of zoobenthos had a positive effect on the food suitability for predator fish. The changing trait compositions of prey influenced the PP linkage-structure, indicating varying specialisation of benthic feeding fish towards available food resources. Our findings suggest that changing morphological characteristics of prey can impact food RS

features for its predators. This approach enables long-term evaluation of prey quality characteristics where no detailed diet data is available and allows for cross-system comparison as it is not relying on taxonomic identities *per se*.

## INTRODUCTION

Community reorganizations, due to natural and anthropogenic pressures on the environment, not only reshuffle species compositions, but also affect species interactions and trophic dynamics (*Tylianakis et al., 2008*). What makes prey species suitable and/or favourable for their predators has been a central question in ecology for over half a century (*Dice, 1949*; *Pyke, Pulliam & Charnov, 1977*) and with accelerating rates of changing communities and shifting interactions (*Harley et al., 2006*; *Both et al., 2009*; *Schmitz & Barton, 2014*), it is still a relevant one.

Besides purely size-structured approaches (*Kerr, 1974*; *Gravel et al., 2013*), the 'who eats whom' is usually described through empirical observations such as gut content analysis (*Hyslop, 1980*; *Hinz, Kröncke & Ehrich, 2005*; *Baker, Buckland & Sheaves, 2014*), stable isotope measurements (*Fry & Sherr, 1989*; *Quillien et al., 2016*), or fatty acid measurements (*Iverson, 2009*; *Kelly & Scheibling, 2012*), resulting in species-based trophic links. This type of data is usually temporally restricted and often limited to commercially relevant species due to high sampling and analysis efforts as well as costs. Although a pure taxonomy-based approach is the fundament of linking predator and prey, it neglects the key ecological information on why a species appears relevant as prey in the first place. Instead of being determined by taxonomic classifications, predation is more likely to be driven by the morphological and behavioural characteristics of a prey species than by its name (*Spitz, Ridoux & Brind'Amour, 2014*). Hence, taxonomic affiliations of species offer only restricted information regarding their suitability as prey or their ecological role in general. During the past decades, the perception of taxonomic limitations in answering ecological questions has led to a steep increase in studies applying different measures linked to functional traits of species, rather than taxonomic related indices, when investigating ecosystem processes or species interactions (*Tilman et al., 1997*; *Cadotte, Carscadden & Mirotchnick, 2011*; *Mouillot et al., 2011*; *Gagic et al., 2015*).

Although it seems straightforward to include morphological and behavioural attributes describing prey, this approach has so far been applied in relatively few studies. Using trait-based approaches in the framework of feeding ecology has contributed significantly to a more complete understanding of predator-prey (PP) relationships and prey selectivity (*De Crespin De Billy & Usseglio-Polatera, 2002*; *Rossberg, Brännström & Dieckmann, 2010*; *Nagelkerke & Rossberg, 2014*; *Rodríguez-Lozano et al., 2016*) through prey vulnerability traits, predator foraging traits, and the interaction of both

prey and predator traits (*Klecka & Boukal, 2013*; *Green & Côté, 2014*; *Spitz, Ridoux & Brind'Amour, 2014*).

In dynamically changing ecosystems, strong environmental drivers, human impacts, and invasive species may rapidly alter community characteristics and biotic interactions (*Collie, Wood & Jeffries, 2008*; *Walther, 2010*; *Johnson et al., 2014*; *Weigel et al., 2015*). Bottom trawl fishing, for example, has been highlighted as one particular human pressure that can impact the composition of benthic prey communities, leading to altered predator conditions by changing the prey availability (*Hiddink et al., 2016*; *Hinz et al., 2017*). Trait-based approaches seem to be a promising tool to assess changing food conditions on community levels (*Weigel, Blenckner & Bonsdorff, 2016*). Yet to our knowledge, there is no available framework to evaluate how changing prey may affect the suitability of food resources of a predator community over time and/or space. Here we suggest a novel concept to assess how changes in prey communities may influence food resource conditions for their predators based on a trait-suitability measure linked to the community weighted mean (CWM) traits of prey. We apply our approach on a marine coastal system in the Baltic Sea, to find out if changes in zoobenthos community structure influenced the food resource suitability (RS) for the benthic feeding fish community. While we focus on the link between zoobenthic prey and benthic-feeding fish in this study, the concept is applicable to all PP systems. The broad spectrum of phenotypic variation in zoobenthos makes it an ideal group to illustrate our approach. For example, many zoobenthos species can be inaccessible for their potential predators as they may be buried too deep in the sediment, have strong physical protection or are not within the right size spectrum to be consumed. In contrast, other zoobenthos may serve as easy prey items for visual hunters when occurring epibenthically, that is, on or just above the sediment surface, and being motile with no physical protection such as hard shells.

The zoobenthic communities in our study system have undergone significant changes in species composition as well as in their functional structure, that is, CWM, over the past four decades (*Weigel, Blenckner & Bonsdorff, 2016*). These long-term changes in zoobenthos CWM indicate significant alterations of the morphological and behavioural characteristics of prey communities from a predator perspective, which could result in altered food resource conditions for benthivorous fish. With currently no other available means to evaluate or predict how the observed functional and structural changes of the zoobenthos community may affect the benthic-feeding fish through changed resource availability, particularly over long temporal scales (in our case several decades), we calculate a trait-based predation suitability measure for coastal fish and examine how changes in CWM of the prey community alter the food RS over space (sheltered and exposed coastal zones) and time (40 years). For building the trait-suitability measure, we first compile information on the prey species spectrum for each predator from the literature. In the following step we translate the species into their trait spectrum, reflecting predation-relevant traits. The frequency of specific traits in the diet results in predation affinity scores, which are ultimately related to the available food resource pool (CWM).

In a system where the link between zoobenthos and fish is strong (*Mattila & Bonsdorff, 1988*; *Snickars, Weigel & Bonsdorff, 2015*), we hypothesise that changes in

Table 1 Benthic-feeding fish predator assemblage (*HELCOM, 2012*; *Snickars, Weigel & Bonsdorff, 2015*).

| Family | Species | Common name | Origin | Warm/cold-water |
|---|---|---|---|---|
| Clupeidea | *Clupea harengus* | Herring | Marine | Cold |
| Cottidae | *Triglopsis quadricornis* | Fourhorn sculpin | Freshwater | Cold |
| Cyprinidae | *Abramis bjoerkna* | Silver bream | Freshwater | Warm |
| Cyprinidae | *Abramis brama* | Bream | Freshwater | Warm |
| Cyprinidae | *Leuciscus idus* | Ide | Freshwater | Warm |
| Cyprinidae | *Rutilus rutilus* | Roach | Freshwater | Warm |
| Osmeridae | *Osmerus eperlanus* | Smelt | Freshwater | Cold |
| Percidae | *Gymnocephalus cernuus* | Ruffe | Freshwater | Warm |
| Percidae | *Perca fluviatilis* | Perch | Freshwater | Warm |
| Pleuronectidae | *Platichthys flesus* | Flounder | Marine | – |
| Salmonidae | *Coregonus lavaretus* | Whitfish | Freshwater | Cold |

the predation-relevant traits of prey communities result in altered RS for the predators. Our aim is to answer whether structural changes of the zoobenthos community, based on predation-relevant morphological and behavioural characteristics, influence the suitability of the food resource pool for the predator assemblage. We evaluate which benthic-feeding fish species are likely to benefit from the functional changes of the zoobenthos and which are more likely to experience lower matching food availability. We further investigate the relationship between the functional diversity of prey communities and the food RS for predators, and lastly, whether altered functional compositions of prey communities may affect the predation linkage-structure.

# MATERIALS AND METHODS

## Study system and predator-prey communities

Our study area comprises coastal zones of the Åland archipelago in the northern Baltic Sea (60°15′N; 19°55′E). The complex land- and seascape form a heterogeneous habitat, encompassing thousands of islands with sheltered bays, exposed open coasts, soft as well as hard substratum, and thus create a multitude of general coastal types. The link between zoobenthos and fish is strong in this area (*Mattila & Bonsdorff, 1988*) as most of the present fish (~95%) are benthivorous during at least part of their life cycle (*Snickars, Weigel & Bonsdorff, 2015*), and thus to a significant part rely on the zoobenthic food resource. In our analysis we included 11 benthic-feeding fish species of the Åland coastal zones, which have been continuously recorded during long-term coastal fish surveys in the study area. Details of the survey method and gear are described by *Snickars, Weigel & Bonsdorff (2015)*. As our study focusses on a qualitative approach to food-resource availability for predators, the continuous presence of included fish species over the past 30 years was set to be a sufficient criterion for their inclusion and relevance in the present study. The fish community includes species of marine as well as freshwater origin, with both classifying either as cold or warm water species (Table 1), which exemplifies our study system as a model area that can be related to marine,

brackish-estuarine, and freshwater systems in environmental settings ranging from sub-arctic (cold water) to boreal (warm) areas.

Zoobenthos communities (prey) included in this work have been monitored over a 40-year time frame from 1973 to 2013 in the same area as the benthic-feeding fish to assure the direct linkage between predator and prey. We included community data from 16 sites comprising two different coastal exposure-zones, sheltered and exposed areas, both representing eight sites. (*Weigel, Blenckner & Bonsdorff, 2015*, *2016*). The exposure-zones provide a proxy for habitat characteristics that have shown to structure the composition and diversity of zoobenthic communities around the Åland Islands, reflecting the exposure to wind and waves, the proximity to land as well as depth (*Weigel et al., 2015*). We included sites that were sampled in 1973, 1989, 2000, and 2013. Each sampling occasion comprised five replicate Ekman-grab samples (289 cm$^2$ per sample) at each site, which were directly fixed in a 4% buffered formaldehyde solution and later identified in the lab to their lowest practical taxonomic unit under a stereo microscope. For further detailed information on the sampling protocol refer to *Weigel et al. (2015)*.

## Trait-based diet spectrum

We selected seven predation-relevant traits for benthivorous fish that reflect quality aspects of zoobenthos as a food resource, covering morphological and behavioural characteristics related to the availability, susceptibility, and palatability of the prey. All seven traits are categorical and contain a total of 24 categories (Table 2). Applied traits were collected from *Törnroos & Bonsdorff (2012)* and *Weigel, Blenckner & Bonsdorff (2016)*. As many fish species are visual hunters, we included one new trait, further characterising zoobenthic prey, indicating if a species is protruding the sediment ('sediment protruding') based on expert judgement.

For the included fish species, we calculated a species-specific trait-based diet spectrum (DS) reflecting the affinity of a predator to specific prey traits and built on the following steps: First, we collected information from studies and open data sources reporting on the prey spectrum for each predator fish species, resulting in data that include the zoobenthos taxa that the predator utilises as prey (Table S1). Based on the presence/absence data of prey species in the predator DS, we built a PP matrix of every predator $i$ and prey $j$ (Fig. 1A), where 1 indicates that predator $i$ is feeding on prey $j$ and 0 indicates that $i$ is not feeding on $j$ (Table S2). Second, we built a prey trait matrix (T), including the selected traits described above (Table 2), for all prey species $j$ and all traits $k$ (Fig. 1B), where 1 indicates that prey species $j$ expresses trait $k$ and 0 indicates that prey species $j$ is not showing trait $k$ (Table S3). The prey trait matrix is built for every predator $i$ and its specific DS. Third, we calculate the predator specific trait DS (Fig. 1C) as the sum of each prey trait category over all prey species in relation to the total number of species in each predator's diet (Fig. 1) with:

$$DS_{i,k} = \frac{\sum_{i=1}^{n} PP_{i,j} \times T_{j,k}}{\sum_{i=1}^{n} PP_{i,j}} \tag{1}$$
**Table 2 List of included functional traits (seven) and trait categories (24).**

| Trait | Category | | Reference |
|---|---|---|---|
| Maximum size | Small | 1–5 mm | [1,2] |
| | Medium | 6–30 mm | |
| | Large | >30 mm | |
| Protection | No protection | | [1] |
| | Tube | | |
| | Burrow | | |
| | Case | | |
| | Soft shell | | |
| | Hard shell | | |
| Fragility | Fragile | | [1] |
| | Intermediate | | |
| | Robust | | |
| Environmental position | Infauna deep | >5 cm | [1] |
| | Infauna middle | within 2–5 cm | |
| | Infauna top | top 2 cm | |
| | Epibenthic | | |
| | Benthic pelagic | | |
| Energy content | Low | <1.7 kJ/g wwt | [2] |
| | Medium | 1.7–3.4 kJ/g wwt | |
| | High | 3.4–5.1 kJ/g wwt | |
| Movement | Swimming | | [1] |
| | Surface crawling | | |
| | Burrowing | | |
| Protruding | Sediment protruding | | [3] |

Notes:
[1] *Törnroos & Bonsdorff (2012)*.
[2] *Weigel, Blenckner & Bonsdorff (2016)*.
[3] Expert judgment.

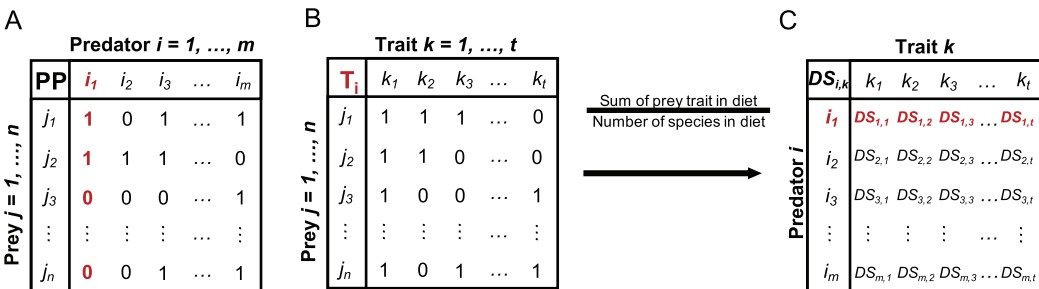

**Figure 1 Conceptual illustration of different steps involved in the calculation of the trait-based diet spectrum.** (A) Starting with binary predator-prey matrix PP with predator species $i$ and prey species $j$, (B) building prey-trait by prey-species matrix T for each predator $i$, (C) calculating trait-based diet spectrum DS for every predator $i$ and prey trait $k$, by relating the sum of each prey trait in predator's diet to the number of species in its diet, as specified in equation 1. Here exemplified with red highlighted steps for predator $i_1$. DS values for trait $k_1 = 1$ would indicate that all prey items of predator $i_1$ express trait $k_1$.

A trait category being present in every prey item of a predator would therefore result in maximum predation affinity of 1, while another trait category being present in only half of the prey items would result in a score of 0.5.

## Food resource suitability proxy

We calculated the potential prey resource availability for predators as the CWMs of trait category expressions. Including 16 sites in every sampling year (1973, 1989, 2000, 2013), all CWM values are based on abundance-weighted traits of zoobenthos, with mean species abundances comprising five replicate samples per site (see *Weigel, Blenckner & Bonsdorff, 2016*). We used the 'FD' library (*Laliberté, Legendre & Shipley, 2014*) in the R environment for statistical computing ver. 3.3.0 (*R Development Core Team, 2013*) to calculate all CWM-values and subsequently standardised them to range between 0 and 1.

Relating the trait-based DS of benthivorous fish to the potential resource pool of zoobenthic prey, we developed a predator-specific proxy for prey RS based on DS and the present prey community structure (CWM traits), at each site and year and for all traits, by calculating the product of the predator specific DS and the trait-based resource availability:

$$RS_{i,k} = \sum_{k=1}^{t} DS_{i,k} * CWM_k \tag{2}$$

with $RS_{i,k}$ being the food resource suitability for predator $i$ and trait $k$, resulting from the predator-specific diet spectrum for prey traits $DS_{i,k}$ as calculated above, and $CWM_k$ being the community-weighted mean of the trait category expression at each site and year. The food resource suitability proxy RS provides a likelihood measure, bound between 0 and 1, for predators to encounter suitable food resources based on the prey traits. Suitability values close to 1 suggest a high prey trait availability for predators in combination with a high affinity in the DS to a particular trait, and therefore results in a high likelihood of the predator encountering the suitable food resource. Values closer to zero reflect a low likelihood when an affinity to a trait in the DS is low and/or the resource availability (CWM) is low.

To test whether the food suitability for the predator community changed over time (1973–2013) and space (sheltered and exposed sites), we built a generalised linear mixed effect model (GLMM) with the resource suitability value RS as response variable (Supplementary Materials, Workflow, and results of GLMM analysis). As fixed factors we included 'year' and 'exposure' and as random factors we included 'sites', 'fish species', and 'traits' to account for non-independence of observation from the same stations, the same predator and traits over time. We analysed the model using the 'nlm4' library (*Bates et al., 2015*) in connection with the 'nlmerTest' library (*Kuznetsova, Brockhoff & Christensen, 2016*) to get significance estimates of model terms. To normalise the distribution of model residuals, all RS values were log transformed ($\log(x + 0.01)$) prior to the analysis and subsequently standardised to zero mean and unit variance.

To shed light on how the food suitability-proxy for each individual fish species varied over time and space, we further analysed species-specific GLMMs, similar to the previous community-based model, with year and exposure as fixed factors and sites

and traits as random factors, to account for non-independencies of observations within each species.

## Functional diversity and food resource suitability

In the present analysis, functional diversity is considered as the diversity in distribution and range of expressed functional traits (*Petchey & Gaston, 2006*), with functional traits reflecting morphological and behavioural characteristics of organisms being relevant from a predatory perspective. We calculated the functional diversity of prey communities as Functional Dispersion (FDis) after *Laliberté & Legendre (2010)*. We chose FDis because it is unaffected by species richness, takes species abundances into account and is capable of handling more traits than species. The metric depicts the abundance-weighted mean distance of individual species to their group centroid (all species of an assemblage) in a multivariate trait-space. FDis was calculated for assemblages at every site sampled (eight sheltered, eight exposed) based on the mean abundances of five samples, using the 'FD' library (*Laliberté, Legendre & Shipley, 2014*). We tested whether the development in FDis changed over time by using a linear mixed-effect model with 'year' as fixed factor and 'site' as random factor, to account for non-independency of observations from the same sites over time. To reveal any relationship between functional diversity and food RS, we fitted a linear model with FDis being the predictor covariate of RS.

## Predation linkage-structure

To find out whether predators show varying specialization and plasticity to the changing trait composition of prey assemblages over time and space, we built PP bipartite interaction networks. Bipartite networks are essentially two-level food webs, in our case linking the zoobenthic prey with the benthivorous predator fish. Besides being applied for studying trophic interaction networks, they are commonly used in pollination webs and seed dispersal studies where every member of the first level is only connected to members of the second level while direct interactions within one level are considered unimportant (*Dormann, Gruber & Fründ, 2008*). Here we used the CWM-values of the zoobenthic assemblages as lower trophic level and connected them with each predator fish, representing the higher trophic level. To account for the affinity of a predator to a prey trait, we weighted each prey-CWM at all stations and for all years with the species-specific diet suitability value in DS. We used two species-level indices to investigate changes in predation linkage-structure over space and time. *Proportional similarity* provides a measure of predator specialisation and describes the dissimilarity between resource use and resource availability, that is, how many of the prey traits are potentially fed on in relation to how many prey traits are present. Values of 1 indicate high generality of predators, where all the available food resources are theoretically also used/ interacted with. Further, we calculated the *sum of interactions per species*, describing the total number of links between predator and food resource. All interaction networks and indices were calculated in the 'bipartite' library (*Dormann, Gruber & Fründ, 2008*; *Dormann et al., 2009*; *Dormann, 2011*) and the R environment.

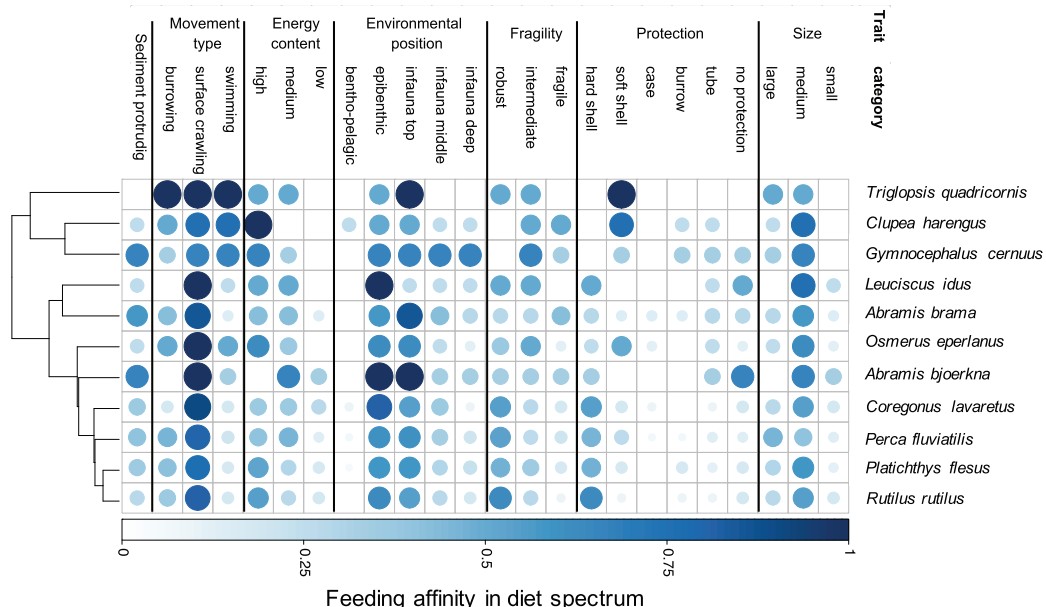

**Figure 2 Trait-based diet spectrum of fish community.** Species are sorted by their similarity in diet spectrum, where similarity-groupings are indicated through the dendrogram. Colour saturation and size indicate the level of trait specific feeding affinity, with '1' indicating trait category is always being preyed on and '0' indicating never being preyed on.

## RESULTS

### Predation profile based on prey traits

Our results demonstrate species-specific feeding spectra of the studied eleven coastal fish species and their affinity towards individual prey characteristics (trait categories) (Fig. 2). Furthermore, our findings indicate that there is a resemblance in DS for a few trait-categories among all fish species, showing a generally high affinity for medium sized, epibenthic and surface crawling prey. However, the majority of traits are more variable regarding their predatory affinity among species. By grouping the fish species according to their DS similarity, we illustrate the relationship in trait-based predation patterns between the predator species (Fig. 2). One close grouping cluster comprises the species *Coregonus lavaretus, Perca fluviatilis, Platichthys flesus, Rutilus rutilus*, showing very similar affinities over the entire trait spectrum. While the four predators utilise all size classes, it is mainly the robust, hard and soft shelled as well as species with no protection, living in the upper part of the sediment and crawling on the surface which they have high affinities to. On the contrary, *Triglopsis quadricornis, Clupea harengus, and Gymnocephalus cernuus* have more specific feeding size classes, not utilizing prey classified as small. All three have no affinity to hard shelled prey. *T. quadricornis* has the highest affinity for soft shelled species and is not utilizing any infaunal prey that is deeper than two cm from the sediment surface (infauna top). It does however have a strong affinity to all prey movement types. *G. cernuus* can utilise a broader range of prey protection types and has equally high affinities to the environmental position of prey ranging from epibenthic to the deep infauna (>5 cm). *C. harengus* shows the highest affinity to

**Table 3 Linear mixed-effect model results and significant approximations for the community model including all 11 fish species.**

| Fixed effects | Parameter estimate | *t*-value | *p*-value |
|---|---|---|---|
| 1973 (Intercept) | −0.0319 | −0.207 | 0.8370 |
| 1989 | −0.0308 | −2.024 | 0.0430* |
| 2000 | 0.0905 | 5.945 | <0.0001** |
| 2013 | 0.1113 | 7.31 | <0.0001** |
| Exposure | −0.0217 | −0.436 | 0.6690 |

Note:
Significance levels are indicated with stars following the significance coding: 0.01 '*' and 0 '**'.

bentho-pelagic prey among all species but only utilises prey with high energy content. *Leuciscus idus* and *Abramis bjoerkna* both are the only predators of the assemblage not showing any affinity for large-sized prey or species with a burrowing movement types. Additionally both predators have the highest affinity to prey with no protection (Fig. 2).

### Food resource suitability over space and time

Time had a significant effect on the overall food suitability for the benthic-feeding predator assemblage (Table 3). Thus, the suitability-proxy calculated as the product of affinity to and availability of food resources (Eq. 2) first decreased towards 1989 (GLMM, $p < 0.05$) and then increased over time in the years 2000 (GLMM, $p < 0.0001$) and 2013 (GLMM, $p < 0.0001$) compared to the values in 1973 (model intercept). There was no effect of exposure on the food quality, implying no significant difference between sheltered, and exposed areas (GLMM, $p = 0.669$) (Table 3).

In accordance with the community model results, the species-specific models show that there is a general trend for most of the fish species (nine out of 11) to experience a significantly positive progression in food suitability over time (Fig. 3, Workflow and results of GLMM analysis). This pattern is gradual with generally highest model estimates in 2013 but shows different species-specific strength. However, there were species for which the food suitability did not improve. For example, food quality for *A. bjoerkna* showed a negative trend until after 2000 and only becomes positive in 2013 while not being significantly different from the suitability measure in 1973 at any point in time (Fig. 3). The food quality for *Leucidus idus* also shows no significant increase and that of *T. quadricornis* decreased in 2013 compared to 2000. Although there is no significant effect of exposure on the food suitability, the model estimates for exposure show both marginal positive, and negative trends for the different species (Fig. 3) (Workflow and results of GLMM analysis).

### Functional diversity—food suitability relationship

The development of functional diversity of prey communities, calculated as FDis, showed an increasing trend over time, with a significant effect of year (GLMM, $p < 0.001$), displaying highest FDis-values and lowest within year variation in the two recent sample occasions (2000 and 2013), indicating a higher functionally diversity compared to the early 1970s and 1980s (Fig. 4A). Furthermore, functional diversity FDis had a highly significant and positive effect on food suitability (lm, $p < 0.0001$, ad. $r^2 = 0.53$) of sampled prey communities (Fig. 4B).

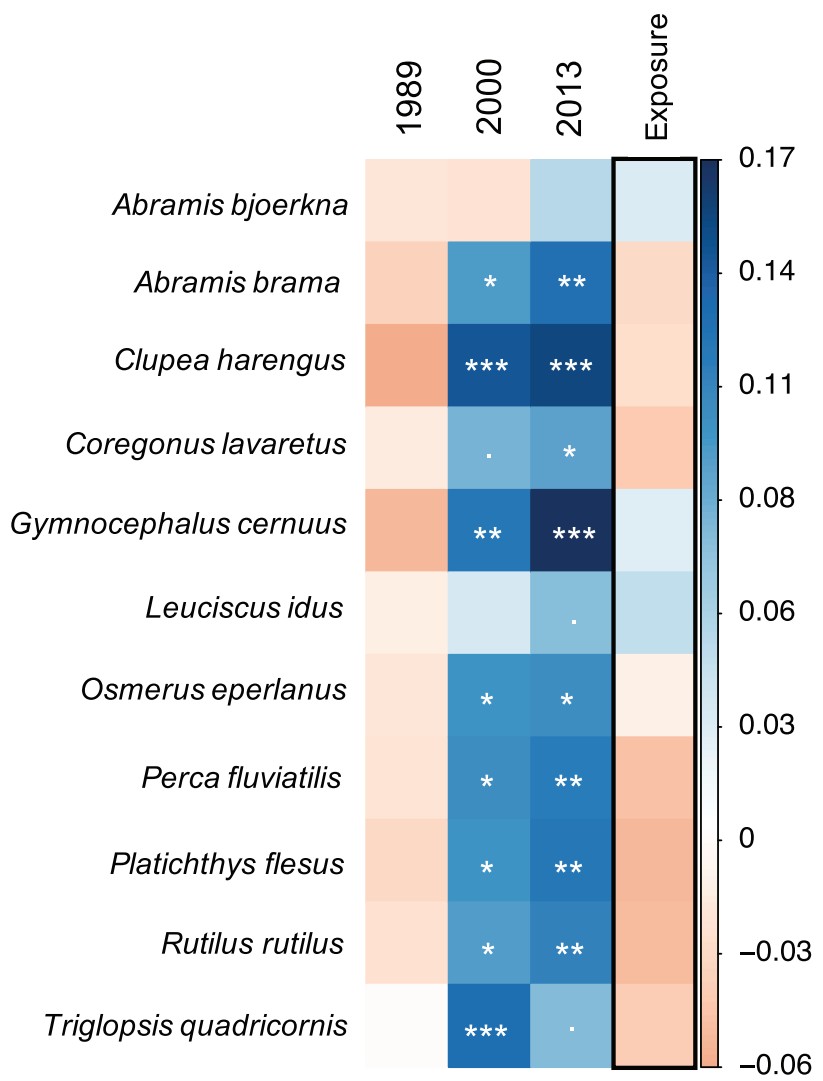

**Figure 3 Species-specific linear mixed-effect model results.** Colours indicate direction of parameter estimates, with blue and orange reflecting positive and negative estimates, respectively. All years are compared against 1973. Significance levels of model parameters for each species are indicated with stars following the significance coding: 0 '***' 0.001 '**' 0.01 '*' 0.05 '.' 0.1 '1'.

## Specificity and resource use of predators

The total *sums of interactions* between predator and prey generally increased over time, especially at exposed sites (Fig. 5A). This implies that the zoobenthos assemblages display a higher number of suitable traits that the predator fish are able to utilise. For all 11 predator fish species, the total *sums of interaction* are the highest in the two later years of the study, 2000 and 2013, a pattern particularly pronounced at exposed sites. Absolute values in the *sums of interaction* at sheltered sites did not change considerably compared to the increasing tendency at exposed sites (Fig. 5A).

The ratio between resource use and resource availability, the *proportional similarity* metric, showed species specificity, revealing the grade of specialisation/generality of the

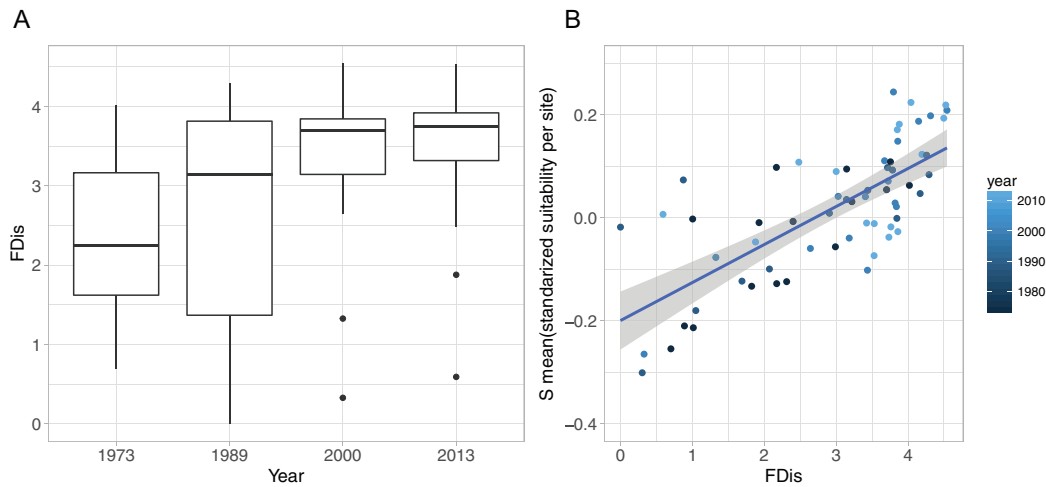

**Figure 4 Functional diversity—food resource suitability relationship.** (A) Development of functional dispersion (FDis) of all sampled prey assemblages ($n = 16$) over time; (B) relationship between food suitability-proxy $S$ and functional dispersion (FDis) of prey communities. Here, $S$ is the mean standardised predictor variable for each prey assemblage at all sites and all sampled years, as applied in the community mixed effect model. Positive relationship is highlighted with a linear model (model estimate = 0.074, $p < 0.0001$, adj. $r^2 = 0.53$).

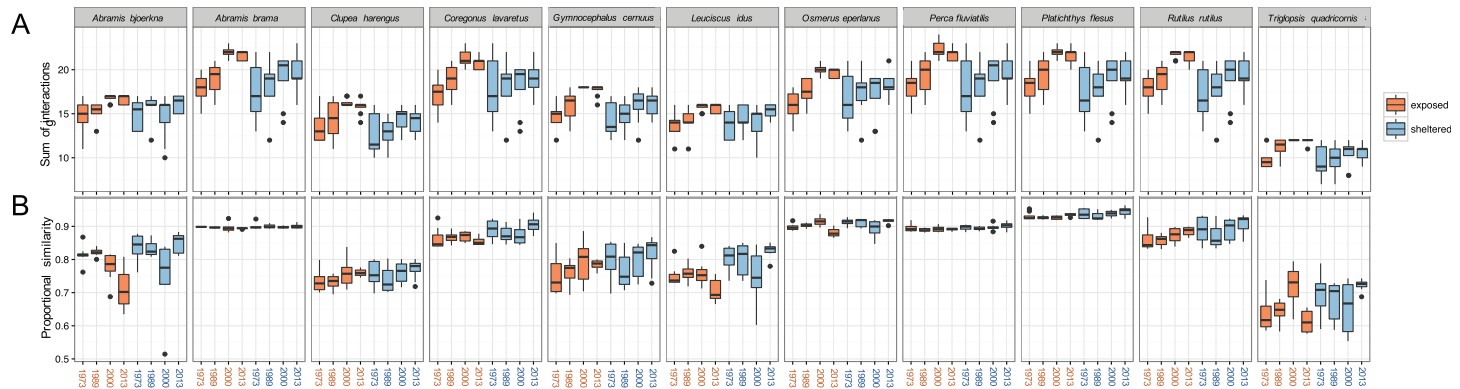

**Figure 5 Predation linkage network structure.** (A) Sum of interactions between predatory fish and the available (trait-based) zoobenthic prey for all sites, (B) 'Proportional similarity' reflecting the ratio between resource use and availability, acting as a measure of specialisation. Each year comprises eight exposed (orange) and eight sheltered (blue) sites.

predator in relation to the present prey community (Fig. 5B). At the exposed sites, *A. bjoerkna* and *L. idus* showed a decrease in *proportional similarity,* implying a higher specialisation towards the available food resource over time. Hence, following the changing prey traits, both predators have less suitable traits to feed on and appear more specialised in relation to the food resource pool. This pattern was, however, not reflected at the sheltered sites. The high generality-values, in combination with minimal within year variation of *A. brama, Osmerus eperlanus, P. fluviatilis,* and *P. flesus,* show a high efficiency in available resource use and a high generality of these predators regardless of the varying *sum of interactions* within years and exposure groups (Fig. 5). The predator community generally displays relatively high values of *proportional similarity* with

*T. quadricornis* being the most specialised species displaying the lowest values overall, which holds particularly true for exposed areas.

## DISCUSSION

### Trait-based predation; concept and food suitability-proxy

Predators select their prey irrespective of taxonomic identities but based on a multitude of phenotypical characteristics, such as morphological, behavioural, and life history traits (*Husseman et al., 2003*; *Klecka & Boukal, 2013*; *Green & Côté, 2014*; *Spitz, Ridoux & Brind'Amour, 2014*; *Rusch et al., 2015*; *Rodríguez-Lozano et al., 2016*). In this study, we applied a novel framework to describe predatory diet suitability based on predation-relevant functional traits of prey. This approach facilitates the understanding on how changes in prey communities may impact the food resource suitability for higher trophic levels, which can potentially result in bottom-up cascades. Our method takes advantage of binary PP links, that is, the presence and absence of feeding links, collected from the literature, to create a trait-based DS that reflects the affinity of a predator towards particular trait categories of a prey species (Figs. 1 and 2).

Following this predator-specific measure, we highlight the composition as well as the theoretical predation-affinity to prey traits (Fig. 2). The comparison of trait feeding affinities among species highlights the similarities and dissimilarities in the food spectra of the predator assemblage. Fish with a similar DS group close to each other. Close groupings of predators, that is, species having similarly affinities to prey traits, such as perch (*Perca fluviatilis*), flounder (*Platichthys flesus*), and roach (*Rutilus rutilus*) (Fig. 2), could serve as an indication for resource competition regarding their zoobenthic prey (*Mattila & Bonsdorff, 1988*, *1989*). This is in line with earlier findings on interspecific competition among these three species (*Persson, 1983*; *Lappalainen et al., 2001*). However, evaluating the impact of competition between the species on the food-resource availability is outside the scope of this study, nonetheless the theory could support future studies considering trait-based measures of interspecific competition.

To evaluate food suitability of a prey community, the measure of feeding affinity to prey traits has to be related to the actual resource availability, that is, the present prey community. As such, CWM traits offer a meaningful link to the PP DS scores, capturing a quantitative measure of the abundance-weighted average community traits. It seems intuitive that high food RS values, that is, high affinity to an available resource, suggests preferable food conditions over low suitability values. However, disproportionate consumption rates in relation to prey abundance are also common, based on dietary preference (*Vadas, 1977*; *Almany & Webster, 2004*), and maximal energetic gain (*Pyke, Pulliam & Charnov, 1977*). In the here applied food suitability measure, predators are assumed to consume the available resources following the proportion of their affinity scores, which includes weighting morphological preferences derived from the observed DS. Yet, the weighing based on presence/absence data does not reflect the different proportions of specific prey items being ingested more or less frequently and potential ontogenic shifts in prey preference are not considered when predator age classes are not taken into account.

Studying a system where long-term structural and functional changes of zoobenthic communities have been described in detail (*Weigel et al., 2015*; *Weigel, Blenckner & Bonsdorff, 2016*), we found that the altered prey CWM had an effect on the food suitability of the predator assemblage over time, confirming our hypothesis. *Weigel, Blenckner & Bonsdorff (2016)* described the major changes in zoobenthic trait composition over time based on community changes in the sheltered and exposed areas we also studied here. Detailing some of their findings and relating them to changes in the here observed food suitability, they found that the changes in sheltered areas between 1989 and 2000 could be linked to decreasing proportions of hard-shell-protected and increasing borrow-protected species, becoming more fragile, increasing in size with higher proportions of large-sized species, and increasing high-energy contents. Similarly, in exposed areas, changes were linked to increased proportions of high-energy species and increasing soft shelled protection. These particular changes in CWM can explain why the two species, *L. idus* and *A. bjoerkna* did not show an increase in food suitability (Fig. 3) as neither displays an affinity for large-sized and soft-shelled species. *A. bjoerkna* additionally showed no affinity for high energy species and *L. idus* none for fragile species, both categories highlighted to have played a significant role in the changing trait composition of the prey communities. Particularly the soft-shelled increase can add to explain the highly significant increase in food suitability for *C. harengus* and *T. quadricornis* which both have high feeding affinities towards this trait. For the period comparing 2000 and 2013 on the other hand, the food suitability for *T. quadricornis* decreased, which can be associated to higher proportions of the prey community being buried deeper in the sediment (*Weigel, Blenckner & Bonsdorff, 2016*). The overall increased food suitability for nine out of the 11 coastal benthic-feeding fish (Fig. 3; Table 3) may have contributed to the increase in total fish abundances (in catch per unit effort) since the mid-late 1990s in the same area despite an overall decrease in zoobenthic biomasses (*Snickars, Weigel & Bonsdorff, 2015*). While our framework centres on a bottom-up approach, aiming to answer how changing prey communities affect predators, it is important to acknowledge that increasing feeding pressures due to higher abundances of predators or changes in the predator species composition may also structure the prey community and its biomass from the top down (*Mattila & Bonsdorff, 1988*; *Olsson, Bergström & Gårdmark, 2013*; *Hinz et al., 2017*). Hence, scenarios would be possible where the predators themselves would decrease the resource suitability by diminishing the CWM of favourable food items.

Against our expectations, there was no significant spatial effect, that is, sheltered or exposed areas, on the prey suitability, although both areas showed significantly different progression pathways regarding the zoobenthic CWM (*Weigel, Blenckner & Bonsdorff, 2016*). This suggests no generality in the effect of changing functional compositions on the suitability-measure *per se* but highlights the importance of particular traits in the food resource, depending on the predator species and its DS (Figs. 2 and 3).

## Functional diversity and prey suitability

Communities with a high functional diversity are assumed to express a wide range of ecosystem functions (*Clark et al., 2012*), supporting resilience towards environmental

change (*Folke et al., 2004*), and promoting the magnitude of ecosystem processes (*De Bello et al., 2010*). In the present study, the functional diversity of prey communities, measured as FDis, increased over time displaying highest absolute values and lowest variation in 2000 and 2013, while the earlier sampling years showed a generally lower absolute FDis and a high within year variation (Fig. 4A). *Weigel, Blenckner & Bonsdorff (2016)* have linked the increased functional diversity to the presence of the non-native and invasive polychaete *Marenzelleria* spp, which seemingly promoted a higher FDis. Although they applied a different set of functional traits, focussing on a broader set of general traits than primarily predation-relevant attributes as in this study, they observed a similar pattern in the progression of FDis. This shows that the FDis pattern remains robust despite the type of traits being applied.

The large within-year variation of FDis can be interpreted as a high spatial discrepancy of prey traits at the assemblage level, with species ranging from a diverse set of different trait combinations at some sites to very similar traits of an assemblage at others (Fig. 4A, 1989). Theory suggests that areas with low functional diversity of prey assemblages should display a higher risk to predators for not finding suitable dietary traits, whereas a high functional diversity of prey would increase the spectrum of available prey traits and thus the chance, for a wider range of predators, to encounter suitable food resources. Hence, a higher functional diversity should support a higher RS. Our results show that elevated FDis of prey communities coincided with an increase in the food suitability for predators (Fig. 4B). This relationship highlights the positive effect of diverse prey-traits on the food resource suitability for predators and supports a positive functional diversity—ecosystem function relationship (*Hooper et al., 2005*).

For specialist species, feeding only on a narrow range of prey items, high functional diversity may be particularly important to ensure matching food items in a rapidly changing environment. Although the benthivorous fish in our study are mostly generalist, they still profit from a higher FDis as the suitable food resource-range increases and therefore the number of possible interactions between predator and prey (Fig. 5A).

Considering the role of the non-native polychaete *Marenzelleria* for the elevated and more stable FDis values (*Weigel, Blenckner & Bonsdorff, 2016*), as well as the observed positive FDis-food suitability relationship (Fig. 4), our results suggest that *Marenzelleria* may feature prey attributes that are generally favourable for the benthivorous fish community. Hence, since its establishment in the early 1990s, *Marenzelleria* seems to act as supporting food item for coastal fish (*Winkler & Debus, 1996*).

A high functional diversity of prey communities supports a broad spectrum of available food resources for predators and could hence serve as a valid approximation for food-quality estimations as depicted in our results (Figs. 3 and 4).

## Feeding specificity and plasticity of predators

The structure of food webs constitutes an important role for ecosystem properties (*Pimm, Lawton & Cohen, 1991*; *Thébault & Loreau, 2003*) and has been linked to the functioning and resilience of communities (*Yen et al., 2016*; *Yletyinen et al., 2016*). Understanding how changes in prey communities affect the predation linkage-structure

of PP networks, can thus provide insight into the resilience of predators to fluctuating food resources, their plasticity to adapt and their efficiency of resource utilisation.

In the present study, we show that an altered functional composition CWM of a prey community may affect the PP network structure and impact the degree of specialisation of a predator towards the available food resource pool (Fig. 5). Although predators showed generally high variability in the total sum of interactions (Fig. 5A), the ability to utilise the present food resources remained stable for most of the predators, and in particular for *A. brama, O. eperlanus, P. fluviatilis,* and *P. flesus* with close to zero variation within and among years and exposure class (Fig. 5B), suggesting a high plasticity in coping with changing prey communities. In contrast, species that show decreasing *proportional similarity* over time, such as *A. bjoerkna* and *L. idus*, primarily in relation to resources in exposed areas, hint towards lower plasticity in resource utilisation of changing prey communities. This finding is also partly reflected in the predator-specific food suitability estimation (Fig. 3), which shows no significant improvement for the respective species over time.

## CONCLUSION

The structural changes in zoobenthos communities around the Åland Islands and over the past 40 years have not only resulted in changes of the species composition but also altered the characteristics of the communities influencing their suitability as prey for the fish assemblage in this area. Based on the changes of prey traits at the community level, we found that the food resource suitability has increased for most of the benthic-feeding predators since the 2000s. Our conceptual framework based on functional traits that reflect predation-relevant morphological and behavioural characteristics of prey can highlight changes in food suitability for predators along changing prey communities. With no other available means to find out if the resource suitability for a predator has changed over long temporal scales, in cases where there is no detailed data on diet, our approach takes advantage of binary PP links, available from the literature, and translates them into trait-based feeding profiles of predators reflecting specific affinities to certain prey traits. These can then be linked to for example, spatiotemporal changes in the functional structure CWM of prey communities and may serve as proxy for the food resource suitability or explanatory factor for PP dynamics. With stomach content data and other cost and sample intensive measurements often lacking, especially on long temporal and broad spatial scales, our approach can help understand altered PP interactions. Being independent of taxonomic species identities, our measure promotes cross-systems comparisons and is applicable to all PP communities. It can further deal with newly introduced and invasive species, as potential food resource, that reflect similar traits of the consumer diet profile.

## ACKNOWLEDGEMENTS

We thank Claire Morandin for helpful discussions during the GLMM development and Malcolm Itter for general comments that improved our study. We are grateful for the constructive comments of two anonymous reviewers that helped improve this study.

### Funding

This paper is a deliverable of the Nordic Centre for Research on Marine Ecosystems and Resources under Climate Change (NorMER), which is funded by the Norden Top-level Research Initiative sub-programme 'Effect Studies and Adaptation to Climate Change', the Doctoral Network Functional Marine Biology (FunMarBio) at Åbo Akademi University, and the Strategic Research Council of the Academy of Finland (grant 312650 to the BlueAdapt consortium). The Finnish Cultural Foundation (Benjamin Weigel) and the Åbo Akademi University Foundation SR as well as the BONUS-Project BIO-C3 also gave financial support (Erik Bonsdorff). The funders had no role in study design, data collection and analysis, decision to publish, or preparation of the manuscript.

### Grant Disclosures

The following grant information was disclosed by the authors:
Nordic Centre for Research on Marine Ecosystems and Resources under Climate Change (NorMER).
Norden Top-level Research Initiative sub-programme 'Effect Studies and Adaptation to Climate Change'.
Doctoral Network Functional Marine Biology (FunMarBio).
Strategic Research Council of the Academy of Finland (grant 312650 to the BlueAdapt consortium).
BONUS-Project BIO-C3 also gave financial support (Erik Bonsdorff).

### Competing Interests

The authors declare that they have no competing interests.

### Author Contributions

- Benjamin Weigel conceived and designed the experiments, performed the experiments, analyzed the data, contributed reagents/materials/analysis tools, prepared figures and/or tables, authored or reviewed drafts of the paper, approved the final draft.
- Erik Bonsdorff contributed reagents/materials/analysis tools, authored or reviewed drafts of the paper, approved the final draft.

### Data Availability

  The raw data are provided in the Supplemental Files.
  Weigel B, Blenckner T, Bonsdorff E (2016) Maintained functional diversity in benthic communities in spite of diverging functional identities. Oikos 125(10): 1421–1433. https://doi.org/10.1111/oik.02894.
  Weigel B, Blenckner T, Bonsdorff E (2015) Data from: Maintained functional diversity in benthic communities in spite of diverging functional identities. Dryad Digital Repository. https://doi.org/10.5061/dryad.6hc8q.

## Supplemental Information

Supplemental information for this article can be found online at http://dx.doi.org/10.7717/peerj.5899#supplemental-information.

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
