# Peer review of "Trait-based predation suitability offers insight into effects of changing prey communities"

_PeerJ, doi:10.7717/peerj.5899_

## Round 0.1 · original submission · Major Revisions

Both reviewers thought your MS a potentially valuable contribution to the literature, but felt the writing needed clarifying on both coarse and fine scales. They have made a large number of thoughtful and constructive suggestions as to how this could be done, in addition to other comments. Please address these in your resubmission.

Reviewer 1 ·

Basic reporting

The manuscript deals with an interesting subject of using knowledge existing knowledge on the prey spectrum of demersal fish from the literature that is generalized through a trait-based approach to investigate changes in invertebrate prey communities through time and its potential effects on the availably of suitable prey for the investigated fish species.

The paper is in parts well written but other parts are very difficult to understand as the authors fail to explain concepts in detail often using language that without additional clarifying sentences cannot be followed. The overall structure of the ms is also difficult to follow. The paper is in essence writtten as a method paper which describes the use of existing knowledge on prey-predator relationships to analyse the potential impact of changes in prey communities in a time series. The reader has however to wait until the method section before she or he can conceptually understand what the papers new method is about. The authors should provide a conceptual description of the method they are introducing already in the introduction. While detail about calculations can go into the method sections there is a need to provide a brief yet complete overview of the methodology they are introducing. This can be followed with the description on what data this method was applied to and what indicators were calculated to describe the specific aims of the study with respect to the long-term data series analysis.

Similarly, the abstract equally lacks a clear language and structure (indicated in general comments). Here the authors explain what can be done with their approach without fully explaining conceptually what their approach entails i.e. taking prey spectrums from published literature, translating them into traits categories to generalize them i.e. eliminating taxonomical concept to describe the suitability of prey communities through time.

The authors should take care to introduce new concepts and to explain those with simple, yet concise language. As an example, the authors talk about “binary predation links” extracted from the literature. Here what they mean is that they used studies reporting on prey spectrums and that they used these to determine the presence or absence of a species/taxa as pey for the fish species forming part of the study. Similar jargon is used throughout the paper and should be avoided. I have tried to highlight as many of these sections as possible in the general comment part of the review.

Overall the literature used is correct although there are important recent papers missing which report on anthropogenic modification of benthic prey communiteis and its effect on fish feeding (Hinz et al. 2017; Scientific reports 7 (1), 6334; Hiddink et al. 2016, Journal of applied ecology 53 (5), 1500-1510; Johnson et al 2014 Proceedings of the Royal Society) as well as older literature (Hinz et al. 2005, Journal of Fish Biology 67, 125-145). With respect to the background this could be enlarged with the mentioned references and others that report about the human induced or natural changes in prey communities and its potential consequences for predatory communities. This aspect is touched upon but within the introduction but could benefit from being supported by examples (one or two sentences would be sufficient here). Also, the abstract should contain the general rational and justification of the study which in general well described in the introduction.

The aim and hypothesis of the paper is not clear. The paper introduces a new method and applies it to a long-term data set. The analysis is then used to interpret the ecological data. Thus, the paper is not an evaluation of the method and it does not aim to show the superiority of the newly introduced methods over traditional species-based prey analyses (at least there is no such section attempting a formal evaluation of the method). I therefore would recommend that the paper clearly states the ecological aims of the paper with respect to the long-term data more specifically. The new method gets introduced to solve ecological questions with the ecological question being the focal point rather than the method. This confusion about what the paper aims to achieve also might have lead to the difficulty in structuring the ms in a way it can be easily read and interpreted.

Experimental design

The paper presents primary research in its scope; however the research questions is not well defined, and the paper appears to be unable to decide if it wants to be a method paper or present an ecological analysis. I appreciate that the authors aimed to do a bit of both, but my recommendation would be to lower the emphasize in both introduction and discussion with respect to the methods capabilities and focus more on the ecological results. The only way the methods can be appreciated is if the results show clear and interpretable results that make sense in an ecological context. Only through presenting a thorough interpretation of the results the reader can evaluate if the method introduce can help interpret trends and provide new insights. In the current form the authors try too hard to sell their idea instead of letting the results speak for themselves. e.g. the abstract, the introduction and other parts of the paper read a lot of: we demonstrate; we evaluate, we show…. novel framework etc.

The method section unfortunately lacks a lot of detail and it is therefore often difficult to follow what precisely has been done. This is often not helped by using jargon or not explaining concepts fully using plain language, one example among others being “binary feeding links” (see above comments). I would recommend that the authors expand on their Figure 1 to also include all other previous steps i.e. dataset using literature data to compile prey spectrums for predators an also include the time series analysis. A schematic flow chart could replace Figure 1 that include all steps of the analysis making it easier for a reader to follow the steps of the data treatment. Also, I would consider labelling the indices in a way within the text that reflects what the abbreviation summarises. species-specific trait-based Diet Spectrum = DS and I would rename S to RS = food Resource Suitability furthermore I would distinguish between RScommunity/species.

The GLMM formulation contains two variables I found difficult to picture in the analysis performed. While I can appreciate that the location is being introduced as a random effect I can not understand why both fish species and traits were introducing as random effects in the model. The identity of species in my mind is irrelevant for this type of analysis and equally the traits. Both were used in the traits approach presented to calculate S for each station and year, why would they subsequently need to be introduce them into the model? The authors need to remove these or explain the need to include these variables as random factors. This applies to both the GLMM performed on the community S and well as on the species S.

Within the functional diversity section, I would recommend that the authors are using the term prey traits diversity, as in my opinion the assumptions about functionality are difficult to justify.

The description on the predation link structure using interaction network analysis is lacking sufficient detail to be understood. I still feel unable to follow what has been done here despite reading the section several times. Equally the results puzzle me; how can the authors conclude anything about prey selectivity with respect to specialization or general feeding and plasticity if they have not analysed the stomach content of fish. All that is being monitored is prey availability changes through time based on literature extracted diet spectrums. Therefore, I cannot follow conclusions such as those put forward e.g. in the abstract “that changing functional compositions of prey can alter the predator-prey linkage-structure by modifying predator specialisation toward available food resources”. What I suspect has been done, is that the authors looked at the number of prey species available within each site and year i.e. potential prey diversity, however, I am not sure due to the language used within this section and the lack of detail to explain concepts. I would recommend rewriting this section using simple plainer language.

The reporting of results is also in parts lacking in detail. Some of the results are loosely summarised without going into the necessary detail representing a lack of technical standard (see general comments). For example, the authors fail to report on the grouping of fish with similar feeding traits in detail. When resporting on the increases in food suitability it is at the beginning not clear what the results refer to the the whole community or individual species. Furthermore there is a lack of bridging sentences or analyses that could have connected the indivdual results reported logically. For example there are several speceis that did not show increasing food suitabilty. It would be interesting to know why these did not respond and what traits were responsible for the lack of response or the reverse. e.g. T. quadriconis appears to have an affinity to soft shell. Such details and depth of anlysis are missing from the ms but would have been useful in providing an in depth understanding of the sytem. This lack of detailed reporting is later also reflected in the discussion which remains at large superficial with respect to the interpretation of the ecological findings of this study (see also general comments).

Validity of the findings

The data presented appears of good quality and the approach used certainly is interesting and has merit. The statistical analysis due to the above mentioned methodological questions could not be fully evaluated. In principle the methods chosen appear sound, but the authors need to address the above mentioned questions. The interpretation of results and conclusions presented could be greatly improved. The interpretation of the ecological results as already mentioned is in my opinion too superficial. As a reader I really wanted to understand the reasons why suitability for some species increased for some species but not for others and which prey traits or trait combinations were responsible. Furthermore, it left me with the question if fish communities in real life responded to these changes, after all the authors present a potential model. Did the fish species for which an increase in prey suitability was detected increase in population size for example or did condition of these improve. I believe only by providing detailed contextual ecological interpretation of the results can the new method introduced be to some extend evaluated and appreciated. In its current form the authors present a new method of data treatment in which the case study is presented as a side story. I would recommend to the authors to rewrite the paper with the focus on the specific ecological question of their case study and to present the method as a tool to get new insights.

I think the paper would also benefit from a section in which the limitation of the introduced approach is discussed. The whole analysis is based on the prey spectrums extracted from the literature for each fish species based one or two existing studies. Studies analysing prey composition from stomach samples can be highly variable and the things ingested at the time when the study was conducted may not reflect necessarily typical prey. Also, often stomach content studies focus on a particular life phase of fish as ontogenetic changes in prey preference can occur. Prey also is found in different proportions in the stomachs indicating prey preferences this has not been considered by this analysis, as no weighting was provided towards preferred prey and only presence and absences were considered. Thus, rare prey items that may be ingested by accident or circumstantially while consuming other prey will have had the same weighting as a dominant frequently ingested prey type. As a minimum requirement I would think that the authors need to discuss these limitations or provide evidence why these are unlikely to influence the results.

Additional comments

The abstract should be restructured as in its current form it’s difficult to follow. I would recommend starting the abstract with the justification of why this study was conducted following the structure of the introduction i.e. why is it important to understand the effect of prey changes to predators considering all the anthropogenic changes occurring today. This should be followed by pointing out the problem of absence of data. After that the method can be introduced in two sentences. Here simply describe the steps you used i.e. prey spectrum from literature, translating prey spectra into trait spectra per species, application of this data to long term changes in benthic communities to establish prey suitability through time and space. Now the application of the method to the specific dataset can be explained. Followed by demonstrating the ecological results. The conclusions should be based on the ecological data with an end sentence justifying through the results the introduced method.

Line 34-35 Not sure how this is possible …..how can predator specialisation be modified if you are using existing literature data and are only exploring the avaialbity of suitable prey??
The introduction is overall ok but from line 87 it become difficult to follow hereafter.

Line 87-94 this section is a bit out of place the information given should form part of the discussion or alternatively you need to work better on an ecological question resulting from your previous results. Thereafter I would recommend starting to introduce your new analytical approach. Less detailed compared to the method section but describing the principle pillars of your approach. Here you have to make sure that you use simple language instead of using jargon. I would follow this up with a clear definition of ecological questions you want to answer. Currently there is a lot of general talk about the method you will introduce and I don’t think this is very useful here.line

Line 151 -160 this section needs more clarity a lot or jargon is used and detail is missing to be understood by an unsuspecting reader. Terms like binary feeding links should be avoided. That these are extracted from past literature needs to be made clear here its not enough just to refer to the supplementary material. I would recommend modifying figure 1 to include all the different data types used and connect those within flow diagram to help the read understand how you treated the data.

Line 163-164 gets in essence repeated in line 172-173. This whole first paragraph I would recommend rewriting as it is difficult to follow

Line 165-167 explain better that CWM is in essence the “potential prey resource avaialbity”.

Line 175 mention the name of the index before you introduce the formula.

Line 186-187 explain what your response variable is I assume its S. I would recommend writing the model you are using if full for both the community and the species level.

Line 188 Why were fish species and traits introduced as random effects?? These were necessary to calculate S so why introduce them here?

Line 198 Why traits as random factor?

Line 200 while the term functional diversity has been around for a while I think unless some functional properties can easily be assumed I would recommend the authors to use the term prey trait diversity. I think it’s more appropriate for the data presented here.

Line 216-227 this section I am afraid I do not understand also after reading and looking at the results I am not sure what is being presented here. What is meant by an interaction network? This section is full of jargon an lacks simple language and detail to be understood.

Line 229-237 This very general summary of the results is not very useful. Please describe results in detail which species did have similar traits and describe groupings. Detail which prey traits where specific to which group of fish.

Line 244 if exposure was not significant it could be removed from the model to give a simpler more robust model.

Line 267-285 I am not able to comment on this section as I could not follow from the method what has been done here. But in general, there appear various parts e.g. line 275 which would belong into the method section. I would recommend rewriting this section in conjunction with the method section so it can be understood by the reader without any prior knowledge or reading required.

Line 290-299 This first paragraph does not add much to the paper and can be removed. Generalized summaries of results are not very useful. The discussion should start with line 301

Line 334 Describe temporal change in the availability of suitable prey which species did benefited and through the increased in which prey traits. I think it would be useful aiming to explain what these changes are based on. There are some explanations in the discussions, but I feel they are currently not concisely presented. Furthermore, as a reader I would like to know have had any effect on fish populations in particular for those species that where you found to have a very significant increase in the avaialbity of suitability prey. Currently there is some very general speculation about an general increase in fish biomass but I think a more complete presentation of results and a more detailed ecological discussion are needed here that include the fish species presented by this study.

Line 340 benthic biomass may decrease due to an increase in fish biomass (i.e. feeding pressure please read Hinz et al 2017 or Hiddink et al. 2016 etc.). As fish populations can limit benthic communities there is an inherent problem with the newly introduced method i.e. prey and potential prey spectrums as analysed here can be modified by through the fish themselves (top down control). This would need to be discussed along with other limitations of this approach.

Line 387-404 unable to comment on this section as I was unable to understand what analysis was performed here.

Line 407-412 I would recommend using the conclusion to summarize your ecological results and to demonstrate that the new method was able to highlight new insights you otherwise would not have attained. Here it is important to clearly name these advantages of your method. In this way readers can evaluate for themselves the usefulness of your approach.

Reviewer 2 ·

Basic reporting

The sentence structure and use of words/grammar needs to be addressed throughout the article. For example I suggest:

Line 18 – should ‘predatory’ be replaced with ‘predator’?
Line 36: ‘on community levels’ be replaced with ‘at the community level’
Line 42: ‘reorganizations of communities’ be replaced with ‘community reorganizations’
Line 57-62: The first sentence is specifically relating to prey suitability and does not lead well to the second one which is more general. Rewrite either one or both sentences and particularly consider what ‘this perception’ is relating to
Line 63: ‘straightforward’ to be one word rather than two.
Line 64: ‘it’ be replaced with ‘this approach’
Line 71: it does not make sense to put ‘especially’ at the beginning of this sentence
Line 82: the ‘while it is applicable to all predatory-prey systems’ does not make sense here. this needs to be reworded
Line 82-86: this sentence needs to be reworded in order to make sense. I also suggest breaking it down into two separate sentences
Line 95: ‘mean’ be replaced with ‘means’
Line 97: ‘especially’ be replaced with ‘particularly’
Line 103: ‘we set out’ be replaced with ‘we aimed’, ‘if’ be replaced by ‘whether’
Line 107-110: reword this sentence as it does not currently make sense.
Line 120: reword the end of the sentence as it does not make sense. Reword the beginning of the sentence to make it clear where the fish were included for example ‘In our analyses we included’
Line 124: ‘of’ be replaced with ‘to’
Line 132-133: ‘of’ be replaced with ‘from’, reword the sentence
Line 134-137: redo the grammar in this sentence
Line 137: ‘have been’ be replaced by ‘were’
Line 153: ‘pray’ be replaced by ‘prey’
Line 186-190: ‘if’ be replaced by ‘whether’, remove ‘has’, put a comma after ‘sites)’, break this sentence into two sentences
Line 195: add ‘varied’ or ‘changed’ after ‘species’
Line 209: ‘sample site’ be replaced with ‘site sampled’
Line 210:’if’ be replaced with ‘whether’
Line 217: ‘build’ be replaced with ‘built’
Line 219: ‘may’ with ‘can’
Line 248: reword sentence as doesn’t make sense
Line 250: ‘instance’ be replaced with ‘example’, ‘shows’ be replaced with ‘showed’
Line 270: spelling for ‘absolute’
Lie 274: remove gap between ‘/ generality’
Line 279-280: add a comma after ‘values’ and after ‘flesus’
Line 283-285: reword this sentence as it doesn’t make sense
Line 289: remove ‘-term’
Line 306-308: reword sentence as doesn’t make sense
Line 318-320: there is repetition between ‘this is in line’ and ‘supporting our claim’, I suggest removing one of these phrases
Line 321: remove the first ‘the’ and consider rewording the sentence
Line 329: ‘compared to its’ be replaced by ‘in relation to prey’
Line 361: reword the sentence as it doesn’t make sense
Line 364: remove ‘the’ and reword the sentence – ‘for predators’ does not make sense in regards to how this sentence is written
Line 374-376: reword the sentence as doesn’t make sense
Line 377-381: reword the sentence – consider breaking it into two separate sentences
Line 393-395: reword the sentence as doesn’t make sense
Figure 2 caption: ‘after’ be replaced with ‘by’

Experimental design

Line 98: please explain what is meant by ‘binary’ when this term is first mentioned

Validity of the findings

Line 290-291: this sentence is repeating the introduction, reword as for discussion
Line 304-306: this sentence is repeating the introduction, reword as for discussion
Line 416: a conclusion should not include something not include any new concepts. In this case neither ‘conservation management’ nor ‘ecosystem assessment tool’ are mentioned previously in the article. If you want to include these here they need to be mentioned in the introduction. They could also possibly be mentioned in the abstract.

Additional comments

I found this research interesting and thorough and believe that it makes a good contribution to its field. However the sentence structure, grammar and use of wording needs to be addressed throughout the article before it is suitable for publication.

---

## Round 0.2 · Minor Revisions

Thanks for making the revisions. Please address the minor points raised by reviewer 2, along with a few additional ones listed below, then I will be pleased to accept your MS.

line 102: 'our study system have'
118: 'hypothesise'
173, 244: hyphenate 'predator-prey'
185: 'would result in'
256: change 'pray' to 'prey'
376: change 'weighing' to 'weighting'
392: change 'they both don't display' to 'neither displays'

Reviewer 2 ·

Basic reporting

Line 91-95: Sentence is too long and therefore difficult to follow.
Line 95: Should ‘however’ be replaced by a different word here?
Line 142-145: This sentence needs rewording e.g. should there be an ‘a’ between ‘as’ and ‘model’?
Lines 346-354: This sentence needs rewording overall, it doesn’t make sense in english.
Line 394 replace ‘non’ with ‘none’.
Line 487: Should ‘changes’ be reworded to ‘changed’.

Experimental design

I am happy with the authors response to reviewers.

Validity of the findings

I am happy with the authors response to reviewers.

Additional comments

Overall I am generally happy with the response to reviewer comments.
However, I still have a few comments, relating to sentence structure/word choice, that I believe need addressing before the manuscript is ready for publishing.

---

## Round 0.3 · accepted · Accept

Thanks for making the final revisions.

#